# Measuring effects of ivermectin-treated cattle on potential malaria vectors in Vietnam: A cluster-randomized trial

**Estee Y. Cramer** [1], **Xuan Quang Nguyen** [2]*, **Jeffrey C. Hertz** [3], **Do Van Nguyen** [2], **Huynh Hong Quang** [2], **Ian H. Mendenhall** [4], **Andrew A. Lover** [1]*

**1** Department of Biostatistics and Epidemiology, School of Public Health and Health Sciences, University of Massachusetts-Amherst, Massachusetts, United States of America, **2** Institute of Malariology-Parasitology & Entomology Quy Nhon, Ministry of Health, Vietnam, **3** U.S. Naval Medical Research Unit TWO, Singapore, **4** Programme in Emerging Infectious Diseases, Duke-NUS Medical School, Singapore

⊕ These authors contributed equally to this work.
* xuanquang_98@yahoo.com (XQN); alover@umass.edu (AAL)

## Abstract

### Background

Malaria elimination using current tools has stalled in many areas. Ivermectin (IVM) is a broad-antiparasitic drug and mosquitocide and has been proposed as a tool for accelerating progress towards malaria elimination. Under laboratory conditions, IVM has been shown to reduce the survival of adult *Anopheles* populations that have fed on IVM-treated mammals. Treating cattle with IVM has been proposed as an important contribution to malaria vector management, however, the impacts of IVM in this One Health use case have been untested in field trials in Southeast Asia.

### Methods

Through a randomized village-based trial, this study quantified the effect of IVM-treated cattle on anopheline populations in treated vs. untreated villages in Central Vietnam. Local zebu cattle in six rural villages were included in this study. In three villages, cattle were treated with IVM at established veterinary dosages, and in three additional villages cattle were left as untreated controls. For the main study outcome, the mosquito populations in all villages were sampled using cattle-baited traps for six nights before, and six nights after a 2-day IVM-administration (intervention) period. Anopheline species were characterized using taxonomic keys. The impact of the intervention was analyzed using a difference-in-differences (DID) approach with generalized estimating equations (with negative binomial distribution and robust errors). This intervention was powered to detect a 50% reduction in total nightly *Anopheles* spp. vector catches from cattle-baited traps. Given the unusual diversity in anopheline populations, exploratory analyses examined taxon-level differences in the ecological population diversity.

**Data Availability Statement:** Data are publicly available at Open Science Framework, https://osf. io/eh7sc/.

**Funding:** This research was financially supported by the Defense Malaria Assistance Program with funds from the Defense Health Agency Research and Development Program (Ref# D1428 to IM). Publication support from UMass (Account SPH-AL-001 to AAL). The funders had no role in study design, data collection and analysis, decision to publish, or preparation of the manuscript.

**Competing interests:** I have read the journal's policy and the authors of this manuscript have the following competing interests: JCH is a military service member of the United States government. This work was prepared as part of his official duties. Title 17 U.S.C. 105 provides that 'copyright protection under this title is not available for any work of the United States Government.' Title 17 U.S.C. 101 defines a U.S. Government work as work prepared by a military service member or employee of the U.S. Government as part of that person's official duties. The views expressed in this article reflect the results of research conducted by the authors and do not necessarily reflect the official policy or position of the Department of the Navy, Defense Health Agency, Department of Defense, nor the United States Government.

## Results

Across the treated villages, 1,112 of 1,523 censused cows (73% overall; range 67% to 83%) were treated with IVM. In both control and treated villages, there was a 30% to 40% decrease in total anophelines captured in the post-intervention period as compared to the pre-intervention period. In the control villages, there were 1,873 captured pre-intervention and 1,079 captured during the post-intervention period. In the treated villages, there were 1,594 captured pre-intervention, and 1,101 captured during the post-intervention period. The difference in differences model analysis comparing total captures between arms was not statistically significant (p = 0.61). Secondary outcomes of vector population diversity found that in three villages (one control and two treatment) Brillouin's index increased, and in three villages (two control and one treatment) Brillouin's index decreased. When examining biodiversity by trapping-night, there were no clear trends in treated or untreated vector populations. Additionally, there were no clear trends when examining the components of biodiversity: richness and evenness.

## Conclusions

The ability of this study to quantify the impacts of IVM treatment was limited due to unexpectedly large spatiotemporal variability in trapping rates; an area-wide decrease in trapping counts across all six villages post-intervention; and potential spillover effects. However, this study provides important data to directly inform future studies in the GMS and beyond for IVM-based vector control.

## Author summary

Malaria incidence in Vietnam has substantially declined in the last decade, however, current prevention strategies may be insufficient to achieve national elimination goals. A proposed tool for use as a mosquitocide is zooprophylaxis-aided ivermectin-based elimination (i.e., treating cattle with ivermectin to kill feeding mosquitoes). Ivermectin (IVM) is an inexpensive helminthicide and is safe for use in mammals. In laboratory studies, mosquitoes feeding on IVM-treated cattle have increased mortality compared to controls. Presented here is a randomized village-based trial to determine whether IVM treatment can reduce the number of captured mosquitoes. Six villages in Central Vietnam were randomly assigned to either receive IVM treatment or remain a control. Mosquitoes in each village were captured for six trap nights, followed by treatment in three villages, followed by trapping for an additional six nights. A difference-in-differences model did not show any statistically significant differences in capture rates between the treated and control villages. Factors including the amount of circulating IVM in the cattle population, crossover of mosquito populations, and differential feeding habits may have affected captures. A total of 18 species were identified using dichotomous keys. In post hoc analyses, no clear trends were observed in the anopheline populations after IVM treatment, as measured by ecological diversity metrics. Future studies should include additional villages, greater control of cattle movements, and consider potential vector movements.

## Introduction

### Malaria transmission in Vietnam

Major progress has been made in many areas toward malaria elimination, and the Greater Mekong Subregion (GMS) is a major focus of these efforts. Through multiple elimination initiatives, malaria incidence in Vietnam has declined substantially over the last two decades; from 2000 to 2019, there has been a 95% reduction in cases and a 96% reduction in malaria mortality [1]. This rapid progress has prompted the Government of Vietnam to set the goal of national *Plasmodium falciparum* elimination by the year 2025, and national malaria elimination due to all *Plasmodium* species by the year 2030 [2].

### Challenges to malaria elimination

Globally, malaria control and elimination programs are focused on high coverage of long-lasting insecticide-treated nets (LLINs), indoor residual spraying (IRS), universal access to artemisinin-based combination therapy (ACT), and rapid diagnostic tests, however, these tools may not be sufficient to achieve malaria elimination in all settings [3,4]. Specifically, progress toward elimination has been stalled due to "residual transmission" which may be driven by combinations of outdoor biting vectors, changes in vector bionomics, and decreased sensitivity to insecticides [5]. These problems are especially complex in the GMS, where a diverse set of vectors have been implicated in transmission, and where peri-domestic vector feeding is common in many settings [6].

### Ivermectin as a tool to accelerate malaria elimination

One promising tool to accelerate malaria elimination is ivermectin (IVM). IVM is an inexpensive and non-toxic helminthicide and mosquitocide with a well-established regulatory environment [7]. And while IVM is lethal to many invertebrates, it has very limited toxicity in mammals [8]. IVM undergoes limited primary metabolism and is excreted largely unchanged after administration in animals.

Other studies with IVM treatment in cattle have been reported, or are underway in a range of settings. Published studies include lab studies in Belize [9], Tanzania [10], Kenya [11,12], and Burkina Faso [13]. Ongoing studies with IVM treatment in cattle and humans include the BOHEMIA study in Mozambique and Kenya [14].

While this drug is practical for use in livestock, its limited half-life necessitates retreatment at regular intervals [15]. Because of the many desirable characteristics of IVM, several prior studies have tested the impact of using this drug to increase mosquito mortality [10,13,16–18]. All studies published to date show that under laboratory conditions, mosquitoes blood-fed on IVM-treated cattle have major decreases in survival rates, especially when fed shortly after drug administration. Several studies have quantified the relative mortality in important vector species in the GMS, including Cramer et al. [18], and Kobylinski et al. [19].

### Literature gaps

Though treating cattle with IVM has been demonstrated to be an effective mosquitocide after being administered to cattle under laboratory conditions, to our knowledge, no field trials have been published examining the impact of IVM treatment in cattle on a village level. By focusing on zoonotic feeding by vectors, a targeted program has the potential to reduce the overall peridomestic anopheline populations in suitable contexts, such as the GMS.

## Entomological context

The GMS (including Vietnam) is an ideal setting for zooprophylaxis-aided IVM-based vector elimination (ZAIVE) as many anopheline species display both zoophilic and anthropophilic behavior in this region [20]. A unique feature of the GMS is the highly diverse range of anopheline species found in the area that are primary and secondary malaria vectors [20]. Incrimination of vectorial capacity is extremely challenging due to both very low sporozoite rates, and the myriad species and vector complexes in the region, leading some experts to state "Given the occurrence of 52 genetic forms in the GMS, of which about 39 forms remain unnamed and their exact species (sensu stricto) are indeterminate, a sensible approach is to delete sensu lato (s.l.) to mean any or all members of the species complex from this point onwards."[21].

The southern parts of the GMS subregion (including Vietnam, Cambodia, and Lao PDR) have some of the highest diversity of potential malaria vectors [22,23]. Vietnam has two major vector complexes (*An. dirus* and *maculatus*), with at least 15 secondary vectors [20,24]. Entomological surveys conducted in the same study villages in Central Vietnam just prior to this study identified both of these 2 primary vector complexes (*An. dirus* and *An. minimus*), "along with 9 secondary malaria vectors: *An. aconitus*, *An. barbirostris s.l.*, *An. harrisoni*, *An. maculatus*, *An. peditaeniatus*, *An. philippinensis*, *An. sawadwongporni*, *An. sinensis*, and *An. vagus.*"[25]. While *An. minimus* and *dirus* are generally found in forested and forest-fringe areas, many secondary vectors are found in peridomestic settings in Vietnam and are a major contributor to residual malaria transmission.

The relative proportions of forest-based, and peri-domestic vector exposures. vary widely by geography, and by season depending on patterns of forest-based activities throughout the GMS; moreover ". . .malaria vectors that preferentially bite outdoors will freely enter these open dwellings and complicate the indoor/outdoor biting distinction" [21]. Moreover, studies in adjacent areas of Cambodia have highlighted indoor biting [26], and a range of studies emphasize the need for continued high coverage of interventions like LLINs in elimination areas due to peridomestic feeding, e.g., Lao PDR, "The findings showed that residual transmission may occur outdoors in the villages, and outside the villages in cultivation fields and forested areas [27]." Finally, control of peridomestic anophelines in villages is essential to ensure "imported" infections can't seed continued transmission. This issue is especially critical in areas with *P. vivax* transmission (a major obstacle to elimination goals) [28] as relapses are generally responsible for many new episodes of parasitemia [29], and where gametocytemia occurs early in the infection. Lastly, building on work by other researchers in Vietnam who proposed the use of cattle baits around human settlements to directly reduce vector density and longevity as a supplement to LLINs, this work uses an "attack and kill" strategy [30].

## Cattle management practices in Central Vietnam

In many rural villages of Central Vietnam, cattle are the major household asset and often serve as dowry. Consequently, livestock health is a major priority, and cattle are treated for any infections including helminths and heartworms, and there is already an established system to administer medication to these animals. However, cattle are not routinely treated with antiparasitics or "cattle dips." In these villages, some families occasionally have a few water buffalos; no other ruminants are present. Typically, cattle owned by a family are allowed to graze freely during the day in areas surrounding these forest-fringe villages, and then are penned adjacent to, or directly underneath, houses at night. Having cattle penned in direct proximity to households provides a clear rationale to directly impact the peridomestic anophelines contributing to the well-documented residual malaria transmission in these villages [25] (Fig 1, photo credit AAL).

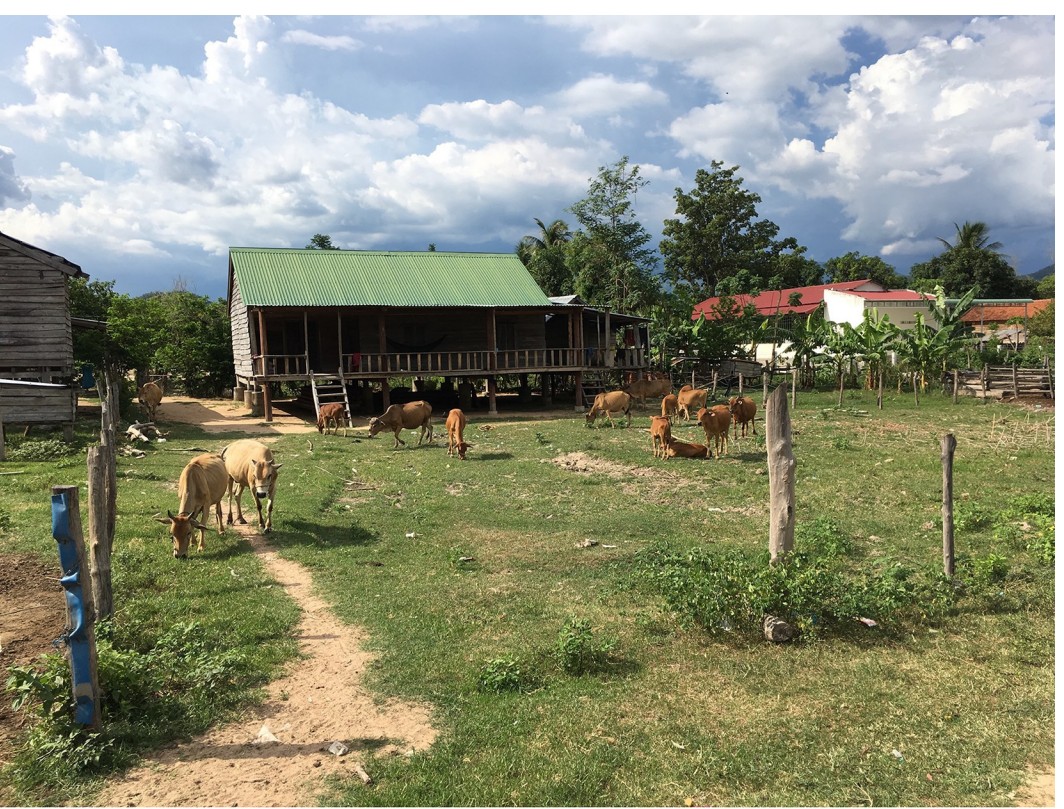

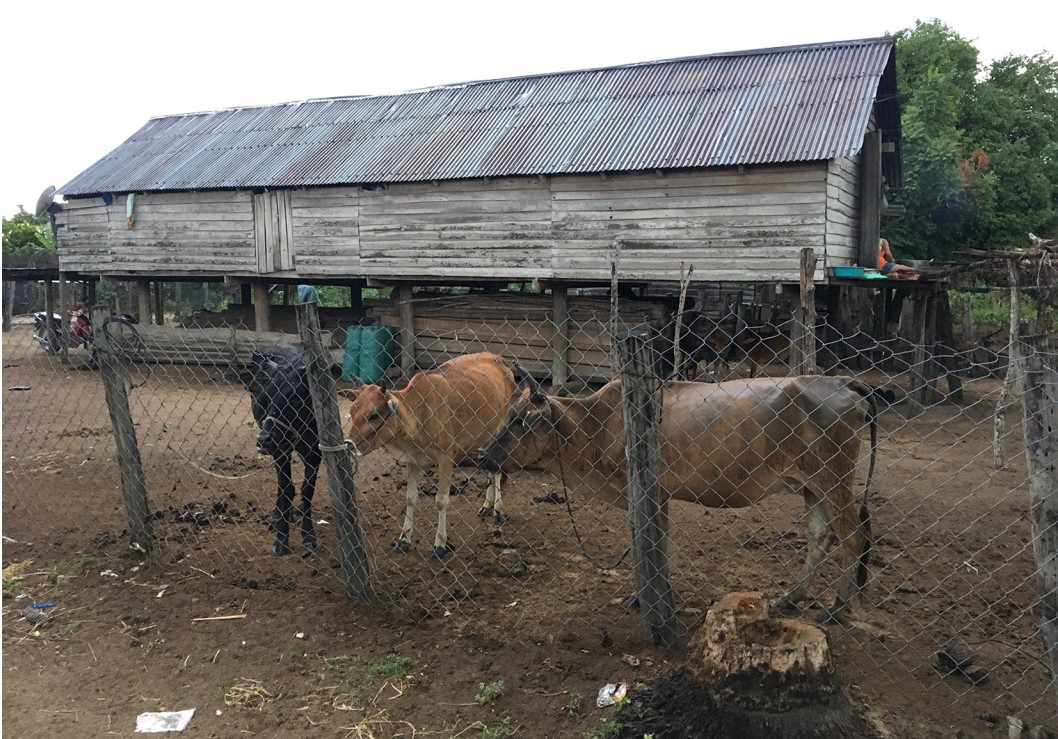

**Fig 1. Representative cattle management in close proximity to households, ZAIVE trial, Central Vietnam, 2019.** (Photographs captured by study authors).

## Study hypothesis

Using a randomized village-based trial, this study quantifies the effect of IVM-treated cattle on the female anopheline populations in treated vs. control villages in rural areas in Central Vietnam. We hypothesize that villages with IVM-treated cattle will have a greater decrease in *Anopheles* mosquitoes captured during the post vs. pre-intervention period as compared to control villages without IVM-treated cattle.

# Materials and methods

## Ethical clearance and consent

IACUC (Institutional Animal Care and Use Committee) approvals were obtained from the National University of Singapore (ref: B18-0303), and the University of Massachusetts Amherst (Approval# 2019–0011); all local regulations were followed. Cattle owners were advised not to sell meat or dairy products for at least 28 days post-intervention. All owners willing to enroll their cattle were informed of these limitations in writing, with follow-up by local Ministry of Agriculture staff to ensure adherence. Verbal consent was obtained from all cattle owners prior to intervention. Treatment of cattle with subcutaneous injections of IVM is routine policy in animal health, and the drug is fully approved in Vietnam for veterinary use (National Guidelines for Animal Health, 2016). No adverse events were reported.

## Study site

This study was conducted in six rural villages in Krông Pa District of Gia Lai province in Central Vietnam [Chính Đơn (CD), Hòa Mỹ (HM), Ơi Jit (OJ), Ơi Đăk (OD), H Yú (HY), and H Lang (HL)] (Fig A in S1 Text). These specific villages were chosen in close consultation with local health staff, who were familiar with the area's cattle-rearing practices and had strong links to local village leaders. The number of individuals living in each village as of 2018 ranges from 325 inhabitants in CD to 1,891 inhabitants in HL. In 2018, malaria was found in all villages except HM; in OD, only 2 malaria cases were reported (API = 6.0 per 1000 population). The largest case burden was reported from CD, with sixteen reported cases in 2018 (AP = 49.2) (Table A in S1 Text).

## Study design

To quantify the relationship between treating cattle with IVM and anopheline mosquito captures, a village-based randomized controlled trial was implemented. In this repeated measures trial, three villages (HM, OJ, and HL) had their cattle treated with IVM, and three villages (CD, OD, and HY) served as controls. The villages were chosen for treatment using a random draw (Stata, *ralloc*) [31].

## Power calculations and sample size

A repeated measures design was used for pre- and post-intervention measurement of outcome; this was essential to maximize study power with the resources available. As stated by Vickers, "...repeating measures can have dramatic effects on power. Increasing the number of follow-up and/or baseline measures from a single one to three or four can reduce sample sizes by 35–70%..." [32]. The trial design was also constrained by the logistics of entomological staff for trapping cycles, and funding available for this pilot study. The number of villages and sampling sites required within each arm for valid statistical comparisons was estimated using a simulation-based approach, as direct analytical methods have not been developed for repeated-measures (longitudinal) experimental designs.

Simulation-based power calculations were used for a range of sampling designs. Simulations were performed using Stata software (version 15, College Station TX, USA); power calculations were performed for a range of villages and sampling events (Table E in S1 Text).

The sample size for the primary evaluation at the end line survey was determined based on the power to detect a difference between treated and control villages, assuming a 50% reduction in total nightly *Anopheles* spp. vector catches from cattle-baited traps. This target reduction was established with consideration of multiple other anopheline vector-control trials (eg, 44% reduction in indoor resting density [33]) and; 50% reduction from ATSBs [34]).

Simulations were parameterized using capture rates from trapping data from the Institute of Malariology, Parasitology, and Entomology (IMPE) (2016; routine public health surveillance trapping), published data collected in Lao PDR [35], and cattle-baited trapping data from Cambodia [36]. Prior trapping data from Krông Pa using CDC traps found a mean of 3.5 vectors/night (median 1; SD = 6.5), and data from Cambodia suggest that cattle-baited traps (CBTs) have nightly capture rates that are 10- to 20-fold higher [36]. With an assumption of a mean of 35 captures per trapping-night, and with a between-village variance of 0.10, to detect a 50% difference in captures with 80% power at a 5% significance level, a total of twelve trapping nights (six pre-intervention and six post-intervention) are required (Table F in S1 Text).

## Cattle census and treatment

Villages were randomized to treatment or control arm; and prior to cattle treatment, a short questionnaire was administered to each head of household to determine the total number of cows that the household owned in treatment villages only. All cows that were pregnant, lactating, currently ill, or under 1 year old were excluded from potential inclusion. After owner consent, all cattle eligible for treatment were injected with a standard veterinary dose of 0.2 mg IVM/kg body weight with a 1% IVM dosage by staff from the local animal health workers using girth-weight charts with validation for Asian breeds [37].

Treatments were conducted across all treated villages over the same two-day period. Veterinary-grade "Vimectin" (Vemedim Corporation; Can Tho, Vietnam) was administered by joint teams from IMPE and local animal health staff. Cattle in control villages were not treated with IVM or a placebo during this study, hence, no blinding was possible.

## Mosquito trapping schedules and logistics

The primary study outcome was a comparison of pre- and post-intervention trapping-night totals of all female anopheline species captured via centralized CBTs (with one trap per village site). These traps were chosen to maximize the study power, as CBTs have been shown to capture 10- to 20-fold more vectors relative to CDC traps in the GMS [36]. and in these study villages specifically [25].

To quantify the impact of IVM treatment on mosquito populations, mosquitoes were trapped every other night over a twelve-day period prior to IVM administration (pre-intervention period). Following baseline mosquito collection, there was a two-day waiting period followed by IVM administration to cattle in the three treatment villages over a two-day period. This was followed by another two-day waiting period, then another round of mosquito trapping (trapping every other night over 12 days) in all six villages (post-intervention period). Overall, a total of 72 trap-site-nights were conducted over the course of this study (twelve nights in six villages).

## Mosquito capture

To capture anophelines, CBTs were used from 18:00 to 6:00 the following morning; traps were swept for mosquitoes using hand aspirators by entomological staff during collection periods

(generally hourly). For each CBT trapping-night, a new cow was randomly selected in each control village. In treated villages, a minimum of four male cows were left untreated, and then each night new random selection was made from the combined total of the four untreated males plus all other untreated cows (pregnant, young calves, etc.). In addition to these CBTs, CDC light traps and double-net traps (DNTs) were used concurrently to capture mosquitoes at these same trapping stations.

## Sample population

The population included in this study was female anopheline vectors captured in CBTs in six villages in Central Vietnam. All captured female *Anopheles* mosquitoes were included in the sample population for this study, regardless of their potential vectorial capacity.

## Mosquito classification

The collections were identified using a standard key to the mosquitoes of Thailand [38]. Identified female mosquitoes were placed into individual cryotubes and stored at -20˚C or -80˚C until processing. Sporozoite rates were not analyzed in this study, as very low indexes are found throughout the GMS (1/1000 or 1/2000) [39], which precludes any valid statistical inferences.

## Statistical analysis

To evaluate the impact of IVM in the treated vs. control villages, a difference-in-differences (DID) model was used. Specifically, differences in trapping totals were quantified between study arms and interventional periods using generalized estimating equations (GEE), with a negative binomial distribution and error-clustering at the village-level. Confidence intervals and p-values for statistical significance were calculated using robust standard errors.

A statistically significant result from this trial was considered to be a p-value $< 0.05$ for the difference-in-differences interaction term (study arm x time period). Main analyses were performed with Stata software (version 16, College Station TX, USA) and other analyses were performed using R version 4.1.2 [40].

In addition to using a negative binomial model with GEE, three additional models were examined as a sensitivity analysis for model specification. These models used GEE with a Poisson distribution and robust errors; a GEE with a negative binomial distribution and bootstraped errors (*xtgee*); and a quantile regression for the median, with robust errors (*qreg*). All additional models were run in Stata.

## Comparisons of anopheline population diversity

To assess the potential impacts of IVM treatment on anopheline species diversity at the village-level, the ecological metric of Brillouin's Index (HB) was used. This index measures biodiversity in collections where it is assumed that there is sampling without replacement, and where one species may be likely to be captured than another (such as mosquitoes being differentially attracted to a CBT) [41,42]. This metric is thus more exact than the more commonly-used Shannon's diversity index; HB is recommended for use in almost all situations in which the aim is to quantify biodiversity. Unlike Shannon's diversity index, p-values to indicate statistical significance are not meaningful for HB. The reported value measures diversity across a sample collection, and therefore has no variance [41]. For HB, the minimum value is zero, which occurs when only one species is present in a population; while the maximum value occurs at: (log N!)/N, where N is the number of individuals in a system. The greater the value,

the greater the biodiversity. Lastly, biodiversity metrics are composed of two components: species richness (a count of the number of species in a population) and species evenness (measured as $p_i \log \log p_i$) where $p_i$ is the relative proportion of a species in a population.

In calculating biodiversity in this sampled population, a set of complementary analyses were performed. First, HB was calculated for each village during the pre-intervention and post-intervention periods. Next, HB was calculated for each individual night of trapping for each arm (treatment vs. control) to assess longitudinal changes in each study arm. Thirdly, HB was calculated for each village for each trapping-night to evaluate changes at the village-level. Finally, the components of biodiversity (species richness and species evenness) were also calculated for each village at each trapping-night, and for each arm at each trapping-night.

## Results

### Treatment coverage

During the intervention period, all eligible and owner-consented cattle in the three treated villages (HM, OJ, and HL) were treated intravenously with IVM at standard dosing. Over two days, a total of 1,112 cattle were treated with 18,761 mL of IVM (1,993 mL HM, 6,200 in OJ, and 10,568 in HL), with IVM dosed at 0.2 mg/kg IVM per total body weight of the cattle.

The arm-level coverage was 73.0% (95% CI: 70.7 to 75.2%; 1112 of 1523). At the village-level, total cattle coverage was over 80% in HM and OJ (81.1% and 83.8% respectively) and 66.8% in HL (Table 1).

### Mosquitoes captured using cattle-baited traps

A total of 5,647 female anophelines were captured using CBTs during the study period; 3,467 were captured before the intervention with 2,180 were captured after the intervention. In the control arm, the greatest number of mosquitoes was captured on trapping-night four; in the treatment arm, the largest number of mosquitoes was captured on trapping-night three. The control arm has the lowest number of captures on trapping-night seven and trapping-night eight; the treatment arm has the lowest number of captures on trapping-night ten. There was less variation in the number of captures in the treatment arm post-intervention in comparison to the control arm post-intervention (Fig 2A). These totals correspond to a mean of 96 per trapsite-night in the pre-intervention phase, and a mean of 61 per trapsite-night post-intervention; these counts are all substantially larger than the minimum required for sufficient study power (Table F in S1 Text).

Across all villages, the greatest number of mosquitoes captured was in CD during the pre-intervention period; 923 mosquitoes were captured, with a median of 141 captured per night. The second largest number captured was in OJ during the pre-intervention period with a total

**Table 1. Ivermectin dosages administered, and treatment coverage, ZAIVE trial, Central Vietnam, 2019.**

| Treated village | Total cattle from census | Total cattle treated | Total ivermectin dosage dispensed (in mL; as 1% solution) | Estimated cattle coverage % (95% CI) |
|---|---|---|---|---|
| Hòa Mỹ (HM) | 148 | 120 | 1,993 | 81.1 (73.8–87.0) |
| Ơi Jit (OJ) | 433 | 363 | 6,200 | 83.8 (80.0–87.2) |
| H Lang (HL) | 942 | 629 | 10,568 | 66.8 (63.7–69.8) |
| total | 1523 | 1,112 | 18,761 | 73.0 (70.7–75.2) |

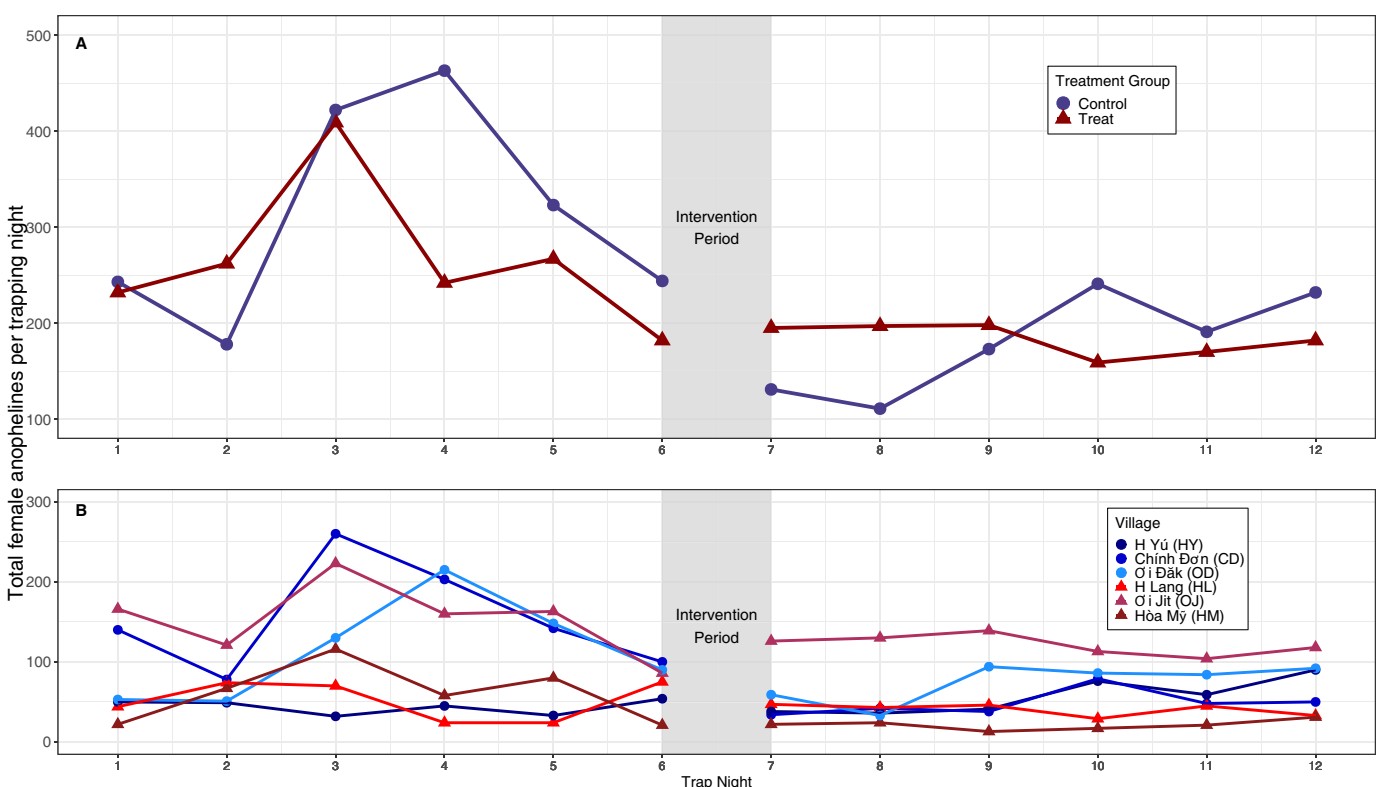

**Fig 2. Total anopheline captures by trapping-night before and after intervention by treatment arm (A) and by village (B), ZAIVE trial, 2019, Central Vietnam.** Number of captured female anophelines in the control arm (blue circles) and the treatment arm (red triangles) over twelve trapping-nights. Values in Panel A represent aggregate mosquito captures across study arm (aggregated treated or controlled villages). Values in Panel B represent counts of mosquitoes captured, by individual village.

of 919 specimens captured, with a median of 161.5 per night (S1 File). In five of six villages, there was a decrease in the total female *Anopheles* captured during the post-intervention period compared to the pre-intervention period. The only exception was the control village, HY, which had a total of 263 mosquitoes captured pre-intervention and a total of 340 mosquitoes captured post-intervention (S1 File and Fig 2B). Across all the sites, eighteen unique *Anopheles* species were morphologically identified. Of these, HY had the highest diversity, with 14 species, CD had 13, HM, OD, and OJ had 12 unique species, while HL had the fewest, with 10 unique species (Fig 2 and Table 2).

**Table 2. *Anopheles* species diversity metrics, ZAIVE trial, Central Vietnam, 2019.**

| Village Name | Study arm | Brillouin's Index, pre-intervention | Brillouin's Index, post-intervention | Total species (pre-) | Total species (post-) |
|---|---|---|---|---|---|
| Chính Đơn (CD) | Control | 1.25 | 1.03 | 13 | 6 |
| H Lang (HL) | Treat | 1.52 | 0.57 | 7 | 7 |
| H Yú (HY) | Control | 1.37 | 1.62 | 11 | 13 |
| Hòa Mỹ (HM) | Treat | 1.15 | 1.02 | 11 | 8 |
| Ơi Đăk (OD) | Control | 1.01 | 0.97 | 11 | 9 |
| Ơi Jit (OJ) | Treat | 1.02 | 1.16 | 11 | 10 |

### Most commonly trapped species in CBTs

The most commonly collected anopheline species across all sites was *Anopheles peditaeniatus* (1,812 captured during the pre-intervention; 1,047 captured post-intervention). The second most commonly captured species were: *An. aconitus* (835 pre- and 522 post-intervention), *An. sinensis* (326 pre and 194 post), and *An. vagus* (242 pre and 113 post) (Fig 3). In CD, HM, OD, and OJ, the number of species captured during the pre- vs post-intervention period decreased. In HL the number of different species remained stable across the study, while in HY there were more unique species captured post-intervention (Table 2, Table B in S1 Text, Table C in S1 Text).

### Mosquitoes captured using human landing catches and CDC light traps

Concurrent with cattle-baited trapping, human landing catches (using a double-net trap) and CDC light traps were also used (18.00 to 06.00) to survey anopheline populations at the same sites. As expected, few female *Anopheles* were captured using these alternative methods. CDC light traps captured a total of 54 mosquitoes during the pre-intervention period, and 69 mosquitoes were captured during the post-intervention period. No anophelines were collected using human landing catches from a double-net trap (Table H in S1 Text).

### Impact of treatment on total mosquito captures (primary outcome)

Of the 18 unique Anopheline species captured, all of which have been reported to be malaria vectors in at least part of their reported range [20,25]. In both the treated and control villages, there was a marked reduction in mosquito captures during the post-intervention period compared to the pre-intervention period. The reduction in the treated villages (30.9%) was less

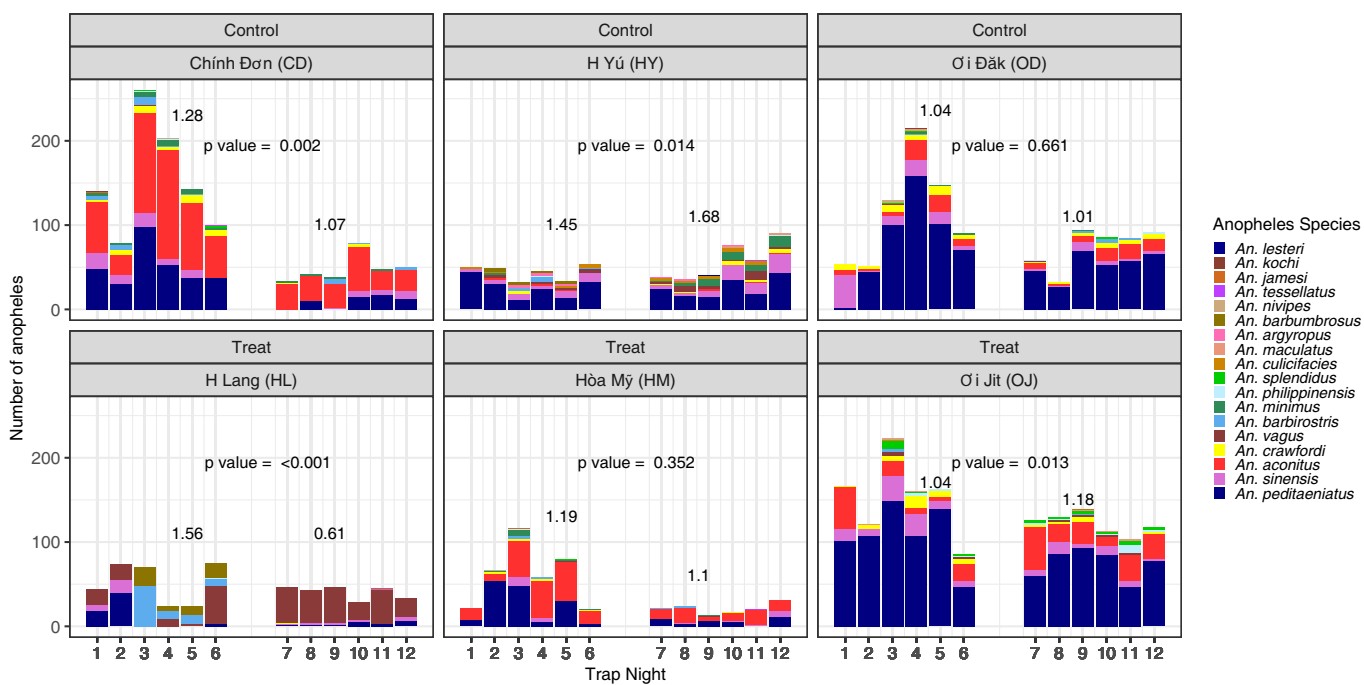

**Fig 3. Count of *Anopheles* species captured nightly, ZAIVE trial, 2019, Central Vietnam.** Count of female *Anopheles* mosquitoes captured with a cattle-baited trap stratified by mosquito species. Values above each treatment and control period are the calculated Brillouin's Index of population diversity.

**Table 3. Results from the primary difference-in-differences analysis, ZAIVE trial, Central Vietnam, 2019.** Results from the generalized estimating equations with a negative binomial distribution. Confidence intervals and p-values were calculated using robust standard errors. The main outcome metric is the interaction term, assessing the difference-in-difference change in mosquito populations due to both time and intervention. The incidence rate ratio and the p-value do not show any statistically significant impacts on the anopheline mosquito captures in the treated groups compared to the control groups across the study periods.

| Comparison | Incidence Rate Ratio (95% CI) | P value |
|---|---|---|
| Intervention to Control arm | 0.85 (0.37–1.97) | 0.71 |
| Post- to Pre-intervention | 0.58 (0.31–1.06) | 0.08 |
| Interaction term | 1.20 (0.61–2.39) | 0.61 |

than the reduction in the control villages (42.4%). Model-based estimates did not provide any evidence for a statistically significant difference in sampled mosquito density between the treatment and control groups, with an interaction term (as incidence rate ratio) of 1.20, (p = 0.61) (Table 3 and Fig E in S1 Text).

Additional statistical analyses (GEE Poisson model with robust errors and negative binomial with bootstrap errors), provided consistent estimates for equation coefficients and confidence intervals (Fig C in S1 Text, Fig D in S1 Text). Importantly, all models had p-values > 0.05 for the interaction term. These sensitivity analyses support the finding of no statistically significant evidence for IVM reducing the number of anopheline captures (Table D in S1 Text).

## Impact of treatment on most common mosquito species captures (secondary outcome)

To determine whether IVM differentially impacted specific anopheline species, additional analyses evaluated whether there were different trends across the six most prevalent mosquito species (Fig F in S1 Text and Table G in S1 Text). Other than *An. peditaeniatus*, a relatively consistent number of captures was found each night after treatment. For *An. peditaeniatus*, there was an initial decrease in captures and then a consistently increasing number of captures for the control arm only (Fig F in S1 Text and S1 File).

## Impact of treatment on Anopheles population diversity (secondary outcomes)

Across study arms, there was no evidence for statistically significant changes in species diversity. However, in each village, there were noticeable changes in the species diversity during the study period. Across the six villages, four experienced a decrease in biodiversity metrics; two control villages and two treatment villages (CD, OD, HL, and HM). Additionally, two villages experienced an increase in measured biodiversity—one control village and one treatment village (HY and OJ) (Fig 3 and Table 2).

When these data were stratified by village, and by trapping-night, there were no apparent trends in trap-night species diversity. The maximum value for the diversity index occurred in three villages (control village CD and treatment villages HL and HM) during the pre-treatment period, and in three villages (control villages HY and OD, and treatment village OJ) during the post-treatment period. In all village-level analyses, there were no clear trends across the timespan of this intervention (Fig 4).

Similarly, there were no clear trends in trapping-night HB values when data was stratified by treatment arm (Fig G in S1 Text). In the control villages, the aggregated day with the lowest measured diversity occurred in the pre-intervention period, while the day with the greatest diversity occurred in the post-intervention period. For the treated villages, the days with the minimum and maximum diversity occurred before treatment administration (Fig 4).

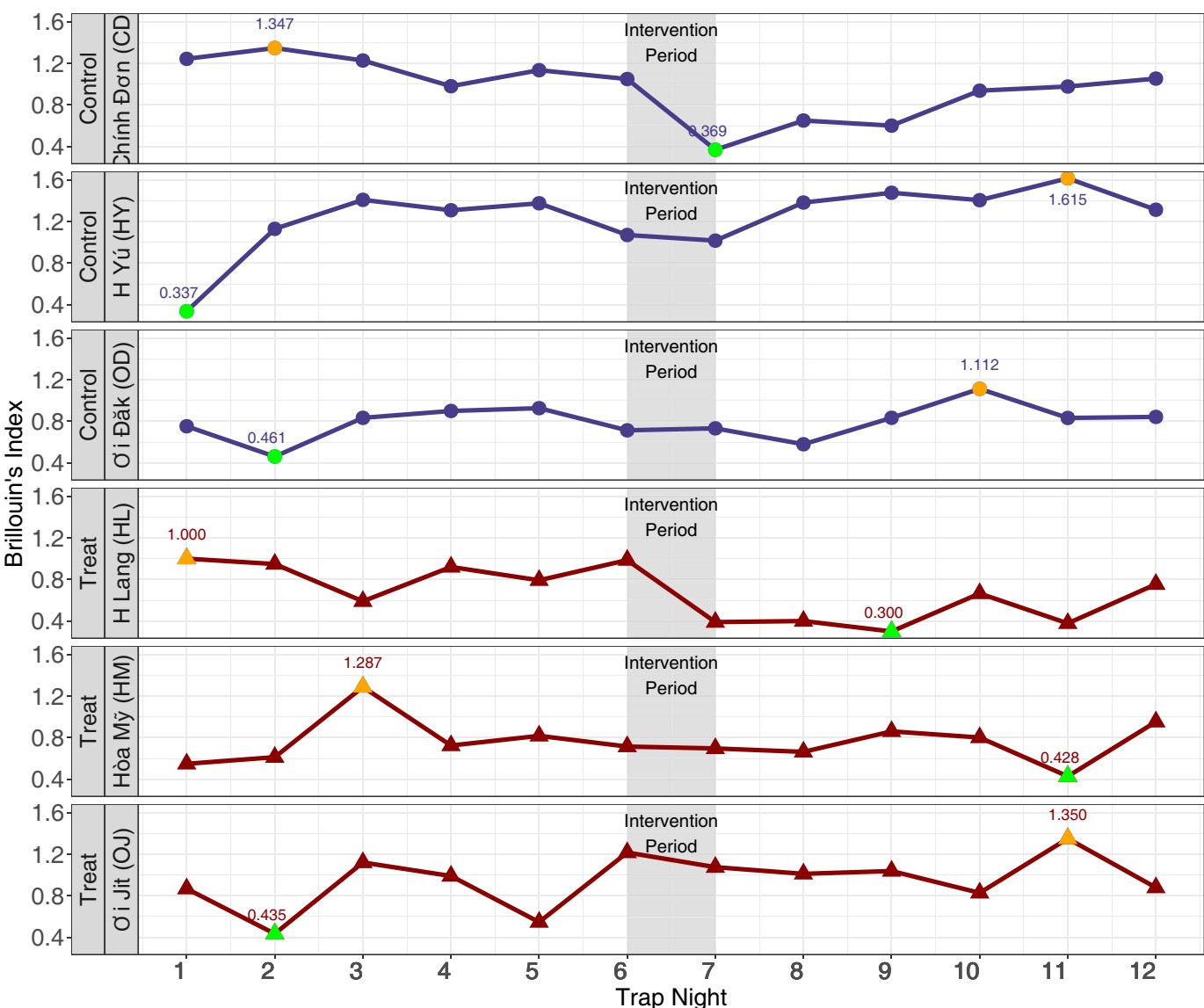

**Fig 4. Brillouin's species diversity index by village by trapping-night, ZAIVE trial, 2019, Central Vietnam.** Brillouin's index was calculated for each trapping-night in each village. Orange markers represent the maximum captured diversity, while green represents the minimum captured diversity.

### Decomposition of Brillouin's Index into richness and evenness

In separate analyses of species evenness and richness, there were no apparent trends across nights post-intervention or across treatment vs. control grouping. Some villages exhibited more species richness during the pre-treatment period and higher species evenness of species evenness during the post-treatment period, however, there are no clear trends over all villages in the treatment or control groups (Fig 5). There also do not appear to be any obvious trends at the arm-level when examining either richness or evenness (Fig G in S1 Text).

## Discussion

This research study was not able to demonstrate statistically significant differences between the study arms as measured by the primary outcome of total anophelines captured in cattle-

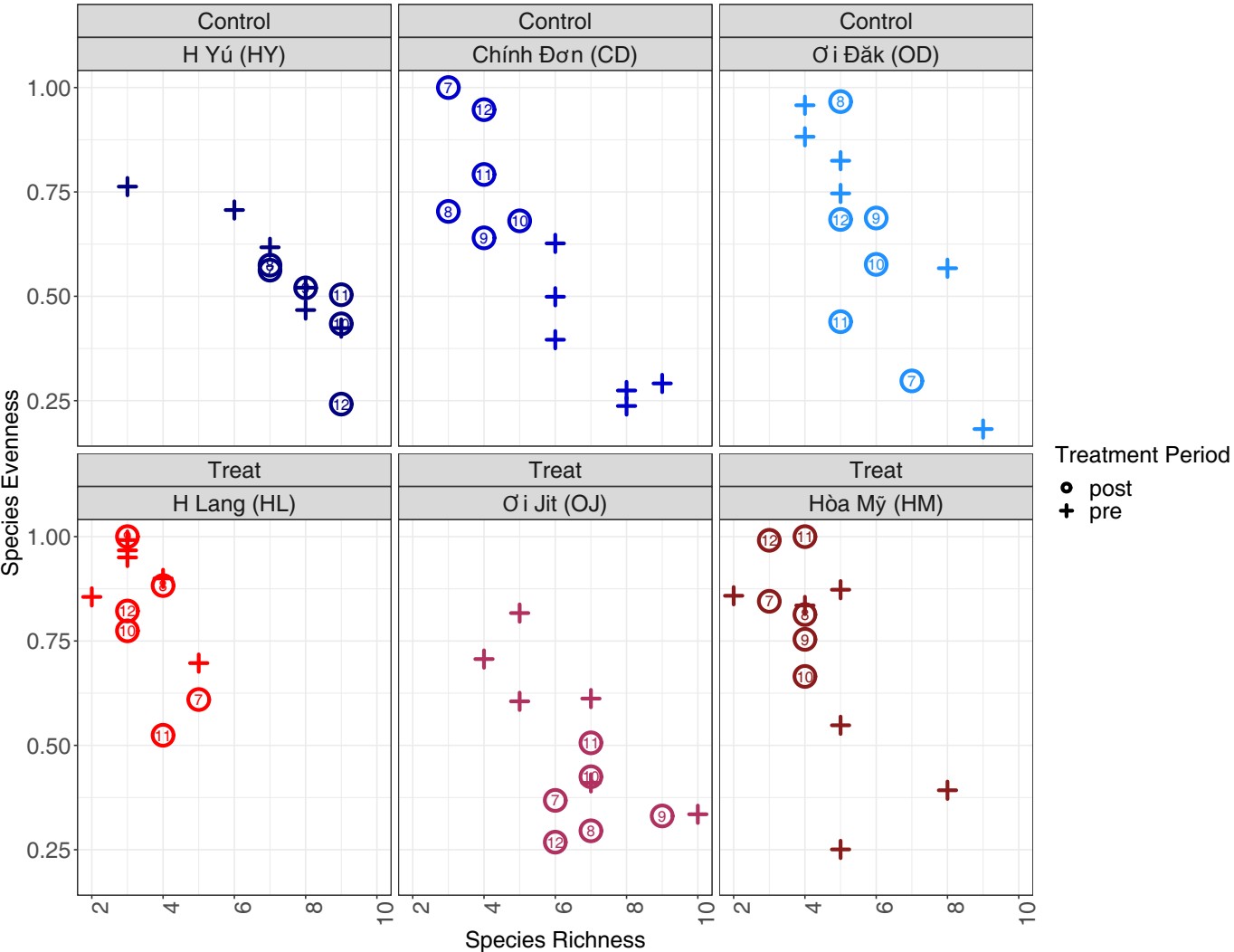

**Fig 5. Village-level trends in anopheline population richness and evenness, ZAIVE trial, 2019, Central Vietnam.** Calculated richness and evenness for each village, by trapping-night. Crosses represent the pre-intervention trap-nights, and open circles represent post-intervention trap-nights. Numbers in circles represent the specific trap-night. Higher values on the y-axis represent increased species diversity, and greater values on the x-axis represent increasing species richness.

baited traps. Ecological diversity indexes were used to assess potential differential mortality in cattle-feeding anopheles. Few species have had quantified IVM-based mortality (Kobylinski [19], and Cramer [18]), and so IC50 values are not known. We postulated that if there were some species more susceptible to IVM-based mortality relative to other species present, the population diversity would show large decreases in the intervention areas when comparing pre- and post-intervention diversity metrics. Conversely, if all species present had generally comparable mortality, no shifts in diversity would be observed.

There were no consistent impacts of IVM on the *Anopheles* species diversity by study arm or by village; four villages exhibited a decrease in diversity while two villages showed greater diversity post-intervention period. Moreover, large decreases in nightly captures were measured during the study across both intervention and control villages. This lack of differential impacts may be due to combinations of factors, as discussed below.

### The total IVM circulating in treated cattle may be insufficient to impact overall anopheline population densities

Evaluating higher cattle coverage levels in areas with larger herds could be important to assess impact "dose density" (the spatial density of treated cows) relative to total anopheline populations with zoophagic or catholic feeding patterns. Alternatively, the use of IVM congeners with higher therapeutic indexes could be an important addition to this intervention so even partial feeds might be lethal to feeding mosquitoes. Lastly, the timeframe of trapping post-treatment in this study was insufficient to measure secondary impacts of IVM (gravity and fecundity) which both lag any direct mortality [43,44].

### There was the potential for crossover in mosquito populations between treated and control villages

The villages in this study population were in proximity to each other: the closest pair were approximately 500m apart. This scenario is supported by the decrease in mosquitoes captured in both the treatment and control villages during the post- vs. pre-period. During a capture-recapture study of *An. maculatus* in Malaysia, 68% of recaptures were taken within a distance of 0.5 km, however, a flight range of 1.6 km was detected [45]. This species is a known vector in adjacent areas of Vietnam [46]. However, the availability of readily accessible blood meals from cattle within each village site suggests this crossover is unlikely.

### Mosquitoes may not have fed on cattle

Many vectors in this population are largely anthropophilic and thus may not readily have fed on the cattle. The most common mosquito species captured in this study was *An. peditaeniatus* which is primarily anthropophilic [47] and therefore is less likely to be impacted by animal-focused interventions. However, as CBTs were used to measure the primary outcome, and human landing catches and CDC traps had very limited capture rates, it appears unlikely that the lack of impact of IVM was due to mosquitoes selecting hosts other than the cattle.

### Spatial spillover

In addition to vector movement between villages, it is possible that the treated cattle may have impacted vectors outside their assigned study arm. This scenario would be important if herd movements coincided with crepuscular feeding, which is common in the GMS [48]. If the cattle are only grazing outside of their randomized area during the day (when few vectors are active) this would be unlikely to impact the outcomes; however, if cattle are grazing outside of their randomized area during dawn or dusk when there is increased vector biting, this would lead to biased results [49]. While spillover effects have had extensive consideration in epidemiology this issue has received limited attention in entomological trials [50]. To account for the lack of clear spatial boundaries caused by cattle grazing in different locations, spatial impacts would need to be included in both the analyses and the design of a future study to account for daily cattle movements and potential vector migration [49,51].

### Impact of moonlight

Animal-baited traps for anophelines have been shown to be greatly impacted by ambient moonlight and lunar cycles [52]. The major peak of trapping (Sept 14 coincided with a full moon) and the nadir was associated with a new moon (Sept 28), results that are well-aligned with prior studies. Treatment and control arms had collections conducted on the same trapping-nights, so there would likely be limited *differential* impacts in trapping rates from

moonlight. However, greatly diminished total captures in the post-intervention may have attenuated our ability to measure any differences in trap-night totals.

Strengths of this study include the fact that it is the first study, to our knowledge, to examine the effect of IVM-treated cattle on wild anopheline populations in a field setting. Though prior work has shown that IVM is an effective mosquitocide in laboratory conditions [18], no prior work has tested the effectiveness in a field trial. The nightly trapping rates greatly exceeded the estimated totals utilized for study design, but this gain was offset by greater-than-expected variability between villages.

Limitations of this study include greater than expected spatiotemporal variability; large decreases in trapping rates across all sites post-intervention; a limited number of villages included in the trial design; and potential spill-over from cattle movements. All of these factors impacted our ability to measure differences in trapping rates in intervention villages. Future studies examining the effectiveness of IVM on a village level should consider including more villages with expansive spatial buffers (e.g., the BOHEMIA trial was designed with 1 km buffers between sites) [14]. While coverage in two villages was above the 80% target, the largest village, and the arm-level coverage was limited at 73%. Future studies should aim to treat at least 80% of the cattle in the village in order to best evaluate reductions in *Anopheline* species; this target is consistent with modeling efforts for IVM-based malaria control, and from prior human MDA campaigns [53].

This pilot study did not find evidence for statistically significant differences in total anopheline captures in IVM-treated villages relative to control villages. The marked decrease in captures across all study sites post-intervention, and unexpectedly large spatial variation in nightly trap rates limited our ability to quantify changes due to IVM treatments. These changes may be due to natural population fluctuations; spill-over of treated cows between villages; or movement of treated vectors. To inform future studies, a range of coefficients of variation from these trial data were used to generate sample sizes for similar studies (Table D in S1 Text).

Future work should prioritize longer-term pre-intervention (pilot) trapping data to fully quantify heterogeneity in trap rates (Fig B in S1 Text) and potential longer-term cycling from lunar phasing; prioritize fully spatially separated clusters despite the increased logical burdens; and consider areas with more constrained cattle movements wherever possible. The use of ecological diversity metrics may also be a useful outcome metric in areas with extremely diverse anopheline populations where vector feeding preferences may be unknown. If IVM treatment is to be an effective tool in reducing vector populations, it is vital to understand how to best scale this intervention for population-level epidemiological impacts.

## Supporting information

**S1 Text.** Fig A in S1 Text. Location of study villages, ZAIVE trial, Sept 2019, Gai Lai Vietnam. (note: C indicates control village; T indicates ivermectin treatment (intervention) village). Table A in S1 Text. Reported malaria cases per study village in 2018, ZAIVE trial, Central Vietnam. Table B in S1. Spatial variation in total anophelines captured via cattle-baited traps (pre-intervention, pooled by study village, N = 6), Sept 2019, Gai Lai Vietnam. Table C in S1. Temporal variation in total anopheline captured via cattle-baited traps pre-intervention, pooled by trap-night, N = 6), Sept 2019, Central Vietnam. Fig B in S1 Text. Coefficient of variation in nightly total anophelines captured via cattle-baited traps (pre-intervention, pooled by trap-night, N = 6), Sept 2019, Central Vietnam. Fig C in S1 Text. Comparison of distribution of trapping-night counts, ZAIVE trial, Sept 2019,

Central Vietnam.

Fig D in S1 Text. GEE model residuals, negative binomial distribution, stratified by treatment arm, ZAIVE trial, Sept 2019, Central Vietnam.

Fig E in S1 Text. Difference-in-differences analysis, using general estimating equations (GEE) model with negative binomial distribution, stratified by treatment arm, ZAIVE trial, Sept 2019, Central Vietnam.

Table D in S1 Text. Alternative model specification for primary outcome, for ZAIVE trial, Sept 2019, Gai Lai Vietnam.

Table E in S1 Text. Study power calculations for pre- and post-interventional designs with entomological outcomes, based on ZAIVE field data. (CV = coefficient of variation). ZAIVE trial, Sept 2019, Gai Lai Vietnam). Table F in S1 Text. Simulation-based study design power calculations, parametrized from IMPE field surveillance data (as in main text), ZAIVE trial, Sept 2019, Gai Lai Vietnam.

Fig F in S1 Text. Total trapping numbers, most common *Anopheles* species per trap-night night before and after intervention; Sept 2019, Gai Lai Vietnam.

Fig G in S1 Text. Brillouin's diversity index calculated for each day of mosquito captures, ZAIVE trial, Sept 2019, Gai Lai Vietnam.

Table G in S1 Text. Differences in pre- and post-intervention trapping totals, by main anopheline species, ZAIVE trial, Sept 2019, Gai Lai Vietnam.

Table H in S1 Text. Trap-night anopheline totals by trapping method, ZAIVE trial, Sept 2019, Gai Lai Vietnam.

Fig H in S1 Text. Decomposition of diversity into richness and evenness components, by study arm, ZAIVE trial, Sept 2019, Gai Lai Vietnam.

(PDF)

**S1 File. File.** Anopheline species captures by date and study village, ZAIVE trial, Sept 2019, Central Vietnam.

(CSV)

## Acknowledgments

We thank all the dedicated staff members at the Institute of Malariology, Parasitology, and Entomology- Quy Nhon for their dedicated partnership throughout this project.

## Author Contributions

**Conceptualization:** Xuan Quang Nguyen, Jeffrey C. Hertz, Do Van Nguyen, Huynh Hong Quang, Ian H. Mendenhall.

**Data curation:** Estee Y. Cramer, Xuan Quang Nguyen, Jeffrey C. Hertz, Do Van Nguyen, Huynh Hong Quang, Ian H. Mendenhall, Andrew A. Lover.

**Formal analysis:** Estee Y. Cramer, Andrew A. Lover.

**Funding acquisition:** Ian H. Mendenhall, Andrew A. Lover.

**Investigation:** Xuan Quang Nguyen, Jeffrey C. Hertz, Do Van Nguyen, Huynh Hong Quang, Ian H. Mendenhall, Andrew A. Lover.

**Methodology:** Xuan Quang Nguyen, Jeffrey C. Hertz, Do Van Nguyen, Huynh Hong Quang, Ian H. Mendenhall, Andrew A. Lover.

**Project administration:** Ian H. Mendenhall, Andrew A. Lover.

**Software:** Estee Y. Cramer, Andrew A. Lover.

**Supervision:** Xuan Quang Nguyen, Huynh Hong Quang, Ian H. Mendenhall.

**Validation:** Andrew A. Lover.

**Visualization:** Estee Y. Cramer, Andrew A. Lover.

**Writing – original draft:** Estee Y. Cramer, Andrew A. Lover.

**Writing – review & editing:** Estee Y. Cramer, Xuan Quang Nguyen, Jeffrey C. Hertz, Do Van Nguyen, Huynh Hong Quang, Ian H. Mendenhall, Andrew A. Lover.

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
