## [Decision Letter · Decision Letter 0]

1 Aug 2023

Dear Dr. Lover,

Thank you very much for submitting your manuscript "Measuring effects of ivermectin-treated cattle on malaria vectors in Vietnam: a village-randomized trial" for consideration at PLOS Neglected Tropical Diseases. As with all papers reviewed by the journal, your manuscript was reviewed by members of the editorial board and by several independent reviewers. In light of the reviews (below this email), we would like to invite the resubmission of a significantly-revised version that takes into account the reviewers' comments. 

We cannot make any decision about publication until we have seen the revised manuscript and your response to the reviewers' comments. Your revised manuscript is also likely to be sent to reviewers for further evaluation.

Sincerely,

Kamala Thriemer

Academic Editor

Nigel Beebe

Section Editor

Reviewer's Responses to Questions

**Key Review Criteria Required for Acceptance?**

**Methods**

-Are the objectives of the study clearly articulated with a clear testable hypothesis stated?

-Is the study design appropriate to address the stated objectives?

-Is the population clearly described and appropriate for the hypothesis being tested?

-Is the sample size sufficient to ensure adequate power to address the hypothesis being tested?

-Were correct statistical analysis used to support conclusions?

-Are there concerns about ethical or regulatory requirements being met?

Reviewer #1: The title of the study is "Measuring effects of ivermectin-treated cattle on malaria vectors in Vietnam: a village randomized trial". My concern is are these mosquitoes really the vectors of human malaria? or do they just happen to bite. All the collections were carried out using cattle bait trap- thus these are mosquitoes that are zoonotic and thus would be responsible for transmission of malaria. I feel at least the title should be changed. There should have been mention of the number of cases in the villages

Reviewer #2: Unsure of the true power of the study. Though it should not impede publication, a more honest discussion in the abstract world help readers understand the frailties.

Reviewer #3: -Are the objectives of the study clearly articulated with a clear testable hypothesis stated? 

Yes, but additional analyses are warranted. Should perform assessment by Anopheles species with enough subjects for analysis.

-Is the study design appropriate to address the stated objectives?

Yes

-Is the population clearly described and appropriate for the hypothesis being tested?

Yes

-Is the sample size sufficient to ensure adequate power to address the hypothesis being tested?

Yes, according to their assumptions. It is not clear where the 50% reduction in Anopheles catches comes from, explain. There should be a section for "Sample Size" separated from the "Study Design". 

-Were correct statistical analysis used to support conclusions?

Yes

-Are there concerns about ethical or regulatory requirements being met?

No

The Methods require more clarification. In particular explanation of the number of collection nights before and after the intervention that the mosquito collections occurred, figures suggest that collections occurred every other night. Was only one cow/location used on each collection night? Also, these are collection nights, do not refer to them as days. What does "swept with hand aspirators" mean, clarify. Clarify what "processing" occurred with the saved mosquitoes? Is this not a cluster randomized trial with the unit of randomization as the village? If yes, then change language, if not, then explain why the "village-based randomized controlled trial" is not a CRT. State the Exclusion Criteria as such, what about lactating cattle?

**Results**

-Does the analysis presented match the analysis plan?

-Are the results clearly and completely presented?

-Are the figures (Tables, Images) of sufficient quality for clarity?

Reviewer #1: The results presented are okay. But would have been useful to have at least a supplementary table showing the numbers for the different species in the various villages before and after.

Reviewer #2: Tables and Figures are clear.

Reviewer #3: -Does the analysis presented match the analysis plan?

Yes

-Are the results clearly and completely presented?

What were the number of cattle for each village, include the control villages as well. Is there human census data? Were there buffalo present in the villages, if so report. What are "units of ivermectin", is this ml, if so explain. 

-Are the figures (Tables, Images) of sufficient quality for clarity?

Dates should not be used in Figures 2-4, instead label each collection night as time before or after the intervention. The numbers in Figure 5 circles do not match the actual collection nights post intervention. Figure 5 is not a common presentation for vector control trials, an explanation of what would be expected impacts of a successful vector intervention on Evenness and Richness. An additional figure should be provided which is an aerial view of the village distribution with scale bars for distance, this will help the reader understand potential spillover concerns better.

**Conclusions**

-Are the conclusions supported by the data presented?

-Are the limitations of analysis clearly described?

-Do the authors discuss how these data can be helpful to advance our understanding of the topic under study?

-Is public health relevance addressed?

Reviewer #1: The authors have mentioned that better results would be obtained if 80% of the cows were treated. However, I feel that malaria cases should have been included to show the effects since most of the Anophelines are not the major vectors.

Reviewer #2: (No Response)

Reviewer #3: -Are the conclusions supported by the data presented?

The duration of IVM activity would have no impact on your analyses. Published results to date suggest one to three effects on mosquito mortality, so how would your 10 day window post intervention be affected by this. You should summarize results of previous cattle treatment studies in the Introduction. LongRange is eprinomectin, not ivermectin, at least cite a long-lasting ivermectin formulation. 

Crossover section would be enhanced if a map of the site was provided. The argument that mosquitoes that ingested ivermectin "need to rest to digest the bloodmeal" is not correct. Sub-lethal ivermectin concentrations reduces locomotor function (Sampaio et al. 2017), and delays time to refeed (Kobylinski et al. 2010, 2017), this argument should be updated. 

The argument of anthropophilic mosquitoes impacting the results begs a breakdown of analysis by Anopheles species. Also discuss the species predilection for feeding on humans or cattle. It is not clear how you would know or not know if mosquitoes were selecting alternative hosts if you were not performing human landing collections.

Previously in paper it was stated cattle kept near the house, so it is not clear how cattle movement would impact in this situation. 

My understanding from your methods is that there is one cow used per trap night per village. If there are 120-629 cattle in the treated village then I see no way that "extensive" trapping of mosquitoes could have occurred. The SolarMal citation specifically placed a density of traps to intentionally reduce mosquito populations. This entire concept of "extensive" mosquito trapping seems highly unlikely and should be removed. 

-Are the limitations of analysis clearly described?

Not really. Major limitations would be a lack of baseline data to understand how abiotic factors such as moon phase impact the mosquito populations. Repeated ivermectin administrations across the duration of the malaria season could help to minimize this issue. 

It is not clear what they mean by "dose-density" it seems they are referring to varying amounts of coverage and not escalating dose levels of ivermectin, this should be clarified. 

-Do the authors discuss how these data can be helpful to advance our understanding of the topic under study?

Yes

-Is public health relevance addressed?

Yes

**Editorial and Data Presentation Modifications?**

Reviewer #1: I find it very difficult to make a decision. I agree a lot of work has gone into the study and thus at least make an attempt to change the title of the manuscript.

Reviewer #2: (No Response)

Reviewer #3: Introduction mentions "peri-domestic vector feeding is common" but most GMS malaria transmission occurs in the forest where the efficient vectors tend to reside, so these points seem rather at odds for malaria control, please explain. 

Ivermectin lipophilicity and metabolisation are two separate concepts. Metabolites have been identified from humans and shown to have mosquito-lethal effect (Kobylinski et al. 2023, Kern et al. 2023) so not sure the point of mentioning these issues? And they should have proper citations. 

Ivermectin half-life when injected into cattle is >2 days so not sure how this is defined as short. 

Citation #12 does not reference mosquitoes. The cattle/Anopheles experiments should be described in better detail, so the reader understands duration of efficacy with a 200 ug/kg dose. 

Why would administration of ivermectin to cattle be ideal for targeting GMS mosquito anthropophilic behavior ?

**Summary and General Comments**

Reviewer #1: Perhaps human landing collection should also have been carried out during this study to indicate if the numbers of mosquitoes coming to bite humans were reduced. This will also provide some comparison as to the different species of mosquitoes biting human/cattle

Reviewer #2: In this paper, the results of the first field trial on the impacts of ivermectin treatment in cattle on mosquitoes are presented. The authors present results on the total number of mosquitoes caught per night across six sampling nights, both before and after the intervention across six villages (3 treatment, 3 control) and also measure changes in species diversity across the study period. The authors do not report a significant difference between the total number of mosquitoes caught between the control and intervention arm. There is a significant difference in the species diversity index between certain villages, but this is not consistent across the intervention/control arms. 

Analysis on number of mosquitoes caught between arms

It is possible that the study design may have impacted the ability to detect a significant difference between the control and intervention arm. The authors describe the methodology used to estimate the power of the study, assuming approximately 35 catches per night from a cattle-baited trap (derived from CBT data in Cambodia) and a 50% reduction in mosquito nightly catches in the intervention arm. This seems very optimistic. The authors have not justified the case for borrowing information on CBTs from the Cambodian setting, so it would be helpful to confirm that similar Anopheles species compositions and feeding behaviours are found in Cambodia. Furthermore, the data presented in Figure 2 shows that the nightly catches in Vietnam often exceeded 35. No justification is provided for assuming that the between village variance was 0.1, the SD of trapping rates being 20 or the distribution used. Given this, and the heterogeneity in monthly catches per night both within and between villages, the study is likely underpowered. This observed low power is often seen in mosquito experiments where conditions can change substantially and should not preclude publication as there is much within the manuscript that is interesting. This is through no fault of the authors, as a lack of baseline data is often a problem, though I do think they were overly optimistic in the assumptions that went into their power calculations. Nevertheless, it is essential that the authors are up front about the actual power the study achieved, so that readers who might not be aware of the problem do not overly interpret results. In such circumstances it is standard practice to calculate the observed power and report it along with the results. This may strengthen the Discussion section of the paper, in terms of ruling-in/out the possibility of the study being underpowered. If the study is underpowered the authors should consider including this within the abstract so that those that just read this know not to interpret the lack of difference between arms as a failure of the intervention. Certainly, I would consider removing the statement " it was well-powered for primary outcomes”" from the discussion, which as I see it has no basis and will confuse interpretation. It would also be helpful if the authors provided the rationale behind their sampling strategy and how it is linked to their understanding of the biological impact of ivermectin treatment in cattle on mosquitoes. The ability to collect mosquitoes by the CBT is influenced by the cattle feeding rate of the mosquitoes and the mosquito life cycle; if mosquitoes are feeding approximately once every three days and certain proportion of these feeds are on cattle, the number of mosquitoes that will be caught in the trap will likely be highly variable. 

The authors used generalised estimating equations (GEEs) to compare the number of mosquitoes caught between the intervention and control arms. However, GEEs are only reliable when there are enough clusters in each arm. Given the heterogeneity in catches within each village (and across arms), the use of GEEs to detect a difference between the intervention and control arm may be limited. Whilst the authors have included error-clustering at the village-level in the model, the small number of villages in the trial may affect the fit of the model to the data. It would be useful to see a figure of the fit of the GEE to the data. The GEE also has a Poisson distri

---

## [Editor Report · Decision Letter 1]

19 Feb 2024

Dear Dr. Lover,

We are pleased to inform you that your manuscript 'Measuring effects of ivermectin-treated cattle on potential malaria vectors in Vietnam: a cluster-randomized trial' has been provisionally accepted for publication in PLOS Neglected Tropical Diseases.

Best regards,

Kamala Thriemer

Academic Editor

Nigel Beebe

Section Editor

<style type="text/css">p.p1 {margin: 0.0px 0.0px 0.0px 0.0px; line-height: 16.0px; font: 14.0px Arial; color: #323333; -webkit-text-stroke: #323333}span.s1 {font-kerning: none

</style>

---

## [Editor Report · Acceptance letter]

22 Apr 2024

Dear Dr. Lover,

We are delighted to inform you that your manuscript, "Measuring effects of ivermectin-treated cattle on potential malaria vectors in Vietnam: a cluster-randomized trial," has been formally accepted for publication in PLOS Neglected Tropical Diseases.

Best regards,

Shaden Kamhawi

co-Editor-in-Chief

Paul Brindley

co-Editor-in-Chief
